# Highly confined epsilon-near-zero and surface phonon polaritons in SrTiO₃ membranes

Ruijuan Xu [1,9], Iris Crassee [2,9], Hans A. Bechtel [3], Yixi Zhou[2,4], Adrien Bercher[2], Lukas Korosec [2], Carl Willem Rischau[2], Jérémie Teyssier [2], Kevin J. Crust[5,6], Yonghun Lee[6,7], Stephanie N. Gilbert Corder [3], Jiarui Li [6,7], Jennifer A. Dionne [8], Harold Y. Hwang [6,7], Alexey B. Kuzmenko [2] ✉ & Yin Liu [1] ✉

Recent theoretical studies have suggested that transition metal perovskite oxide membranes can enable surface phonon polaritons in the infrared range with low loss and much stronger subwavelength confinement than bulk crystals. Such modes, however, have not been experimentally observed so far. Here, using a combination of far-field Fourier-transform infrared (FTIR) spectroscopy and near-field synchrotron infrared nanospectroscopy (SINS) imaging, we study the phonon polaritons in a 100 nm thick freestanding crystalline membrane of SrTiO₃ transferred on metallic and dielectric substrates. We observe a symmetric-antisymmetric mode splitting giving rise to epsilon-near-zero and Berreman modes as well as highly confined (by a factor of 10) propagating phonon polaritons, both of which result from the deep-subwavelength thickness of the membranes. Theoretical modeling based on the analytical finite-dipole model and numerical finite-difference methods fully corroborate the experimental results. Our work reveals the potential of oxide membranes as a promising platform for infrared photonics and polaritonics.

Surface phonon polaritons (SPhPs) are hybrid optical modes bound to interfaces between optically distinct media, resulting from the coupling of photons with optical phonons in polar materials. They exist within the Reststrahlen bands, the spectral regions (typically in the far- and mid-infrared range) between transverse optical (TO) and longitudinal optical (LO) phonon frequencies where the real part of the optical permittivity is negative[1]. Typically, SPhPs have a weak confinement in bulk crystals, with a wavelength close to that in free space. When the thickness of the crystal—or membrane—is decreased to the deep-subwavelength scale, SPhP modes at each interface hybridize to form two branches, with symmetric and antisymmetric distributions of the normal component of the electromagnetic field with respect to the membrane center. The antisymmetric mode is lower in energy and is propagating with a momentum significantly larger than in electromagnetic waves of the same frequency in free space; the symmetric mode is pushed up in energy and approaches the LO frequency (where the permittivity is zero) when the membrane thickness is small enough, in which case it is referred to as the epsilon-near-zero (ENZ) mode[2,3]. The dispersion of the ENZ mode is almost flat in momentum, and it extends inside the light cone, where it becomes a radiative

[1]Department of Materials Science and Engineering, North Carolina State University, Raleigh, NC 27606, USA. [2]Department of Quantum Matter Physics, University of Geneva, 1211 Geneva, Switzerland. [3]Advanced Light Source Division, Lawrence Berkeley National Laboratory, Berkeley, CA 94720, USA. [4]Beijing Key Laboratory of Nano-Photonics and Nano-Structure (NPNS), Department of Physics, Capital Normal University, Beijing, China. [5]Department of Physics, Stanford University, Stanford, CA 94305, USA. [6]Stanford Institute for Materials and Energy Sciences, SLAC National Accelerator Laboratory, Menlo Park, CA 94025, USA. [7]Department of Applied Physics, Stanford University, Stanford, CA 94305, USA. [8]Department of Materials Science and Engineering, Stanford University, Stanford, CA 94305, USA. [9]These authors contributed equally: Ruijuan Xu, Iris Crassee. ✉e-mail: Alexey.Kuzmenko@unige.ch; yliu292@ncsu.edu

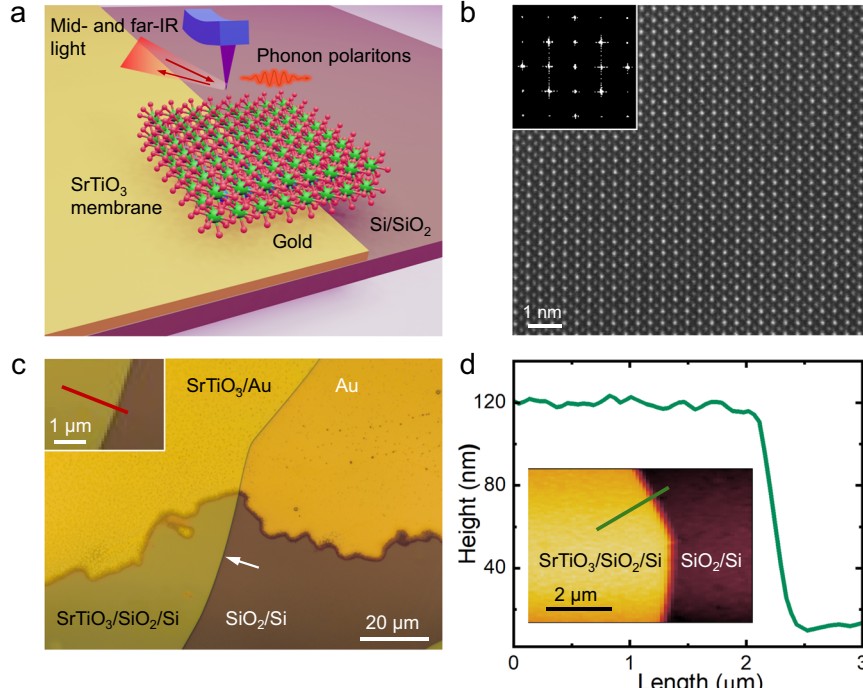

**Fig. 1 | Preparation and structural characterization of SrTiO₃ membranes.**
**a** Schematics of the s-SNOM/SINS measurement on an SrTiO₃ membrane. **b** Atomic-resolution STEM imaging of a freestanding SrTiO₃ membrane suspended on a SiNₓ TEM grid. Inset shows the Fourier-transform of the STEM image. **c** The optical image of the SrTiO₃ membrane transferred on a thermally oxidized Si substrate with part of the surface coated by a 50 nm thick gold film. The white arrow denotes the edge of the membrane. The inset shows the trace of SINS line scan across the edge of the membrane supported on SiO₂/Si substrate. **d** A line scan of the height profile across the edge of the membrane. The inset indicates the trace of the line scan.

Berreman mode[3–5]. Importantly, both modes enhance the electromagnetic field on scales much smaller than the diffraction limit, and thus show great promise for infrared nanophotonic applications such as sensing, perfect absorption, superlensing, optical switching, nonlinear optics, coherent thermal emission, and thermal management[4–14].

To realize these applications, transferable, scalable single-crystalline thin-film materials exhibiting strong optically active low-loss phonon modes are indispensable. Some quasi-two dimensional van der Waals (vdW) compounds, such as hBN[6], GeS[7], GaSe[8] and α-MoO₃[9,10] exhibit phonon polaritons (PhPs) with low-loss and deep-subwavelength confinement, and their remarkable figures of merit have advanced various nanophotonic technologies in the infrared region[11–13]. These materials, however, are typically available as manually exfoliated flakes with a lateral size on the micrometer scale, limiting the scalability for device fabrication. Additionally, the omnipresent optical anisotropy of the vdW materials, although sometimes beneficial for observing some exotic optical phenomena, such as hyperbolic PhPs, may in certain cases be an unwanted complication. Furthermore, the choice of Reststrahlen bands in the vdW crystals is quite limited, which is a factor inhibiting potential applications. Thus, it is important to examine different material families offering isotropic bands in other spectral regions. Cubic and pseudocubic perovskite oxides exhibit optically intense phonon modes and also support low-loss PhPs[14–17]. In particular, strontium titanate (SrTiO₃) stands out as one of the most technologically developed and broadly used materials in oxide electronics and is a host of many exciting physical phenomena, including incipient ferroelectricity[18], dilute superconductivity[19] and interfacial 2D electron gas[20]. In addition, SrTiO₃ exhibits tunable phononic and photonic properties through electrical and optical excitations, strain control, and controlling the concentration of oxygen vacancies and chemical dopants[20–25]. Recent advances in the synthesis of free-standing, large-scale crystalline oxide membranes with a thickness close to the unit-cell limit have provided new opportunities for

polaritonics and photonics[26–30]. For instance, theoretical studies have suggested the presence of highly confined SPhPs with a good propagation quality in ultrathin SrTiO₃ and other perovskite membranes down to the monolayer limit[31]. Such phonon polaritons, however, have not been experimentally studied so far.

Here, we explore SrTiO₃ membranes as a new promising platform for PhPs in the infrared regime. Combining far-field Fourier-transform infrared (FTIR) spectroscopy and near-field synchrotron infrared nanoscopy (SINS), we experimentally confirm both the antisymmetric and symmetric SPhP modes, including the radiative Berreman mode, in a 100 nm crystalline SrTiO₃ membrane transferred onto a thermally oxidized silicon substrate, part of which is covered by gold (Fig. 1a). At this thickness, which is below one percent of the free-space wavelength, the symmetric mode is a truly ENZ mode, with a giant enhancement of the electromagnetic field inside the sample. Moreover, via nanoscopic broadband SINS imaging near the sample edges, we reveal propagating antisymmetric modes exhibiting a momentum 10 times larger than SPhPs of the same energy in bulk SrTiO₃. The experimental results match very well with our calculated near-field scattering signals based on the extended-dipole modeling and finite-difference computations.

## Results

### Synthesis and transfer of SrTiO₃ membranes

We prepare an epitaxial heterostructure of 100 nm SrTiO₃ thin films with a 16 nm Sr₂CaAl₂O₆ water-soluble sacrificial buffer layer synthesized on single-crystalline SrTiO₃ substrates by pulsed-laser deposition (Methods and Supplementary Fig. 1). X-ray diffraction performed on the as-grown heterostructure, reveals that the films are epitaxial, single-phase, and high crystalline quality (Supplementary Fig. 2). After dissolving Sr₂CaAl₂O₆ in deionized water, a millimeter-scale SrTiO₃ film is released from the substrate of growth and transferred onto a SiO₂/Si substrate. Part of the surface of the latter is coated by 50 nm

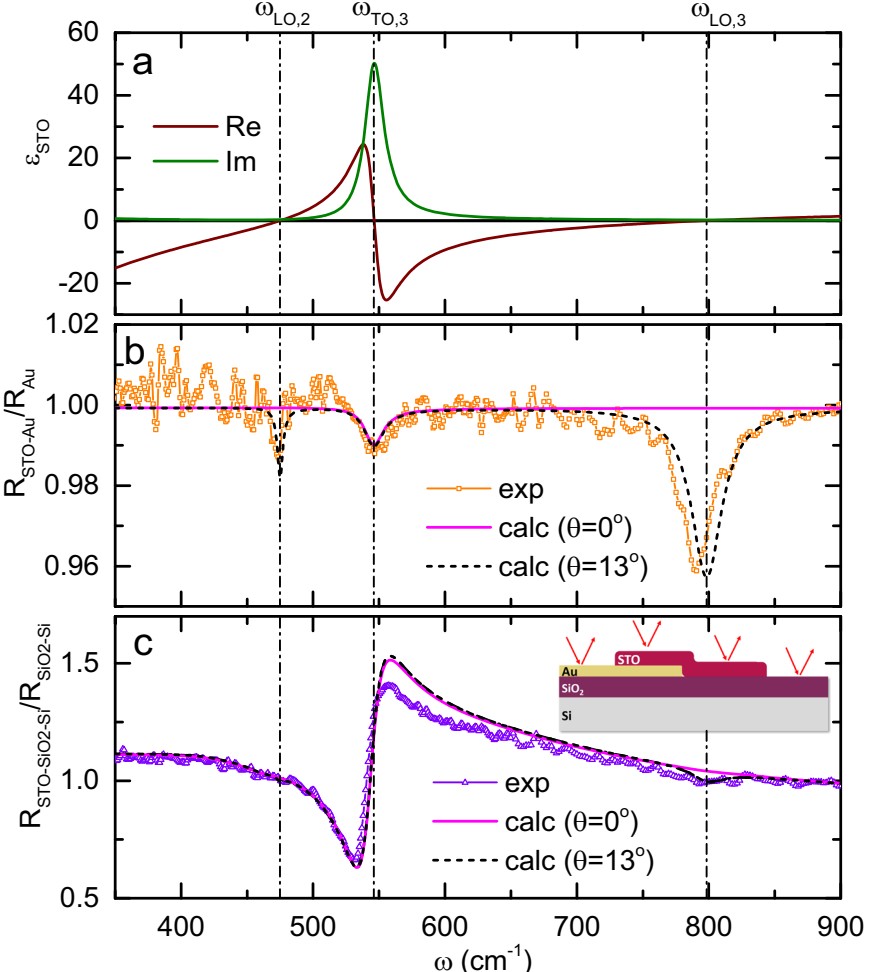

**Fig. 2 | Analysis of far-field reflectivity spectra of the SrTiO₃ membrane. a** The real and imaginary parts of the dielectric function of SrTiO₃ at room temperature obtained by a factorized-formula fitting of the normal-incidence reflectivity measured on bulk SrTiO₃. **b** Far-field reflectivity of a 100 nm SrTiO₃ membrane laminated on gold (symbols) normalized to the reflectivity of bare gold, and calculated spectra using the dielectric function of bulk SrTiO₃ for normal incidence (solid line) and for $\theta = 13°$ (dashed line). **c** Far-field reflectivity of the same SrTiO₃ membrane on SiO₂/Si normalized to the reflectivity of substrate (symbols), and calculated spectra using the dielectric function of bulk SrTiO₃ for normal incidence (solid line) and at $\theta = 13°$ (dashed line). Vertical lines in all panels refer to LO and TO frequencies of bulk SrTiO₃.

thick gold (Fig. 1a, c), allowing us to directly compare the PhP properties within the same sample on top of a metal and an insulator. The resulting membrane is free of cracks and wrinkles over an area of several hundred micrometers and of a high crystalline quality, which is retained after the lift-off, as revealed by atomic-resolution scanning transmission electron microscopy (STEM) (Fig. 1b). The AFM topography (Fig. 1d) indicates that the membrane surface is atomically smooth with the RMS roughness of ~850 pm and that the edge is sharp within the resolution limited by lateral tip dimensions.

### Far-field infrared reflectivity and Berreman modes of SrTiO₃ membranes

As a reference for the quantitative analysis of far- and near-field spectral data presented in this paper, we use the dielectric function $\varepsilon_{STO}(\omega)$ of bulk SrTiO₃ (solid lines in Fig. 2a) as a function of frequency $\omega$. We obtain it by measuring the far-field FTIR reflectivity spectrum at near-normal angle of incidence on a bulk SrTiO₃ crystal and fitting it using a factorized formula with three phonon oscillators $\varepsilon_{STO}(\omega) = \varepsilon_\infty \times \prod_{i=1}^{3} \frac{\omega_{LO,i}^2 - \omega^2 - i\gamma_{LO,i}\omega}{\omega_{TO,i}^2 - \omega^2 - i\gamma_{TO,i}\omega}$, where $\omega_{LO,i}$ and $\omega_{TO,i}$ are the LO and TO frequencies, $\gamma_{LO,i}$ and $\gamma_{TO,i}$ are the corresponding scattering rates and $\varepsilon_\infty$ is the high-frequency dielectric constant ( see Supplementary

Note 1 and Supplementary Fig. 3). The three modes relevant for this paper, $\omega_{LO,2} = 475 \pm 0.3$ cm⁻¹, $\omega_{TO,3} = 546.1 \pm 0.8$ cm⁻¹ and $\omega_{LO,3} = 798.4 \pm 1.7$ cm⁻¹, are shown in Fig. 2a as vertical lines (parameters of other modes are given in Supplementary Table 1).

Next, we analyze the FTIR reflectivity spectra of the SrTiO₃ membrane on different substrates obtained using an infrared microscope (see Methods). Figure 2b presents the reflectivity of the membrane laminated on Au, normalized to the reflectivity of bare gold. The reflectivity is close to one since the sample is much thinner than the optical penetration depth (Supplementary Fig. 4). Nevertheless, one can clearly see three dips at about 475, 545 and 790 cm⁻¹, which are close to $\omega_{LO,2}$, $\omega_{TO,3}$ and $\omega_{LO,3}$ respectively (the one at $\omega_{LO,2}$ is weak but still distinguishable above the noise level, see also Supplementary Note 2 and Supplementary Fig. 5). To understand the origin of the dips, we first calculate the reflectivity of the membrane at normal incidence ($\theta = 0$) (solid line in Fig. 2b). It reproduces the $\omega_{TO,3}$ dip, which is due to the usual optical absorption, but expectedly does not capture the LO ones because of the transverse nature of light. Second, we calculate the reflectivity at a small but finite angle of incidence ($\theta = 13°$), which is consistent with the numerical aperture of the focusing objective (black dashed line in Fig. 2b). This time, the simulation mimics both LO dips

remarkably well. The absence of the LO structures in the zero-angle simulation suggests that they appear due to the coupling to the z-axis (normal to the surface) component of the electric field of light. To corroborate this interpretation, we perform the same measurements using two objectives with different numerical apertures and therefore different average angles of incidence (Supplementary Fig. 5), and we clearly observe the enhancement of the LO phonon structures with the increase of $\theta$. Based on this evidence, we identify both dips at $\omega_{LO,2}$ and $\omega_{LO,3}$ as the Berreman modes[3,5,32,33], where the electric field is almost fully trapped inside the sample and oscillates perpendicular to the surface. Figure 2c presents the reflectivity (symbols) of the same membrane laminated on SiO$_2$/Si, normalized to the reflectivity of bare substrate (See Supplementary Note 3, Supplementary Fig. 6, and Supplementary Table 2 for the analysis of reflectivity and dielectric functions of SiO$_2$/Si substrate). The corresponding calculations at normal incidence and for $\theta = 13°$ (solid and dashed lines respectively) agree well with the experiment. As the insulating substrate screens the electromagnetic field much weaker than gold, the TO mode in Fig. 2c is much more prominent than in Fig. 2b. On the other hand, the Berreman mode at $\omega_{LO,3}$ is significantly less pronounced on SiO$_2$/Si than on gold and the one at $\omega_{LO,2}$ is hidden below the noise level.

A closer look at the experimental and model curves in Fig. 2b and c reveals a certain difference between the actual phonon mode frequencies in the membrane and in the bulk SrTiO$_3$ sample. Direct least-squares fitting of the experimental curves in Fig. 2b and c (see Supplementary Note 4 and Supplementary Fig. 7) provides us the values of the frequency and the scattering rates for these modes (Supplementary Table 3). Notably, a red shift of 2.8, 3.4, and 8 cm$^{-1}$ is found respectively for the modes $\omega_{LO,2}$, $\omega_{TO,3}$ and $\omega_{LO,3}$ in the membrane as compared to the bulk, which is beyond the error bars. This interesting fact is possibly a strain- or surface- related effect, which deserves a separate study.

From the same fitting (Supplementary Table 3), we learn that the two Berreman modes have linewidths of $6.3 \pm 1.7$ cm$^{-1}$ and $33.1 \pm 2.3$ cm$^{-1}$, respectively, matching well the scattering rates $\gamma_{LO,2} = 5.0 \pm 0.5$, cm$^{-1}$ and $\gamma_{LO,3} = 27.7 \pm 2.3$, cm$^{-1}$ in bulk SrTiO$_3$ (Supplementary Table 1). This gives rise to quality factors $Q = \omega/\gamma$ of about 60 and 30 for the two longitudinal modes respectively. Such significant Q-factors result from the small values (<0.3) of the $Im(\varepsilon_{STO})$ at $\omega = \omega_{LO,2}$ and $\omega_{LO3}$, which are significantly lower than the values for plasmonic ENZ materials including indium tin oxide (ITO) and metals[4,34,35].

## Near-field spectroscopy of ENZ and surface phonon polaritons
Further insights into the nature of the phonon polaritons in the STO membrane can be inferred from scattering-type scanning near-field optical microscopy (s-SNOM), which has shown its potential for studies of ultrathin, two-dimensional, and nanostructured materials[6,9,36,37]. In our work, we perform synchrotron infrared nanospectroscopy, where broadband synchrotron IR light is focused onto the apex of a metal-coated tip of an AFM operating in tapping mode. Scattering of light on the sharp tip provides the necessary momentum for the optical excitation of the phonon polaritons[38]. By scanning the tip over the sample, s-SNOM amplitude ($s$) and phase ($\phi$) spectra are independently measured as a function of the light frequency and the tip position $x$ and demodulated at several harmonics of the tapping frequency (Methods). Figure 3a (symbols) shows the second-harmonics amplitude spectrum $s_2(\omega)$ obtained on the gold-supported SrTiO$_3$ membrane far away from the sample edges and normalized to signal on bare gold as sketched in Fig. 3e. The data feature two outstanding asymmetric resonant peaks, just below the frequencies $\omega_{LO,2}$ and $\omega_{LO,3}$ respectively. These structures are remarkably intense, presenting a stark contrast with only a 4% percent reduction in the far-field reflectivity due to the Berreman modes (Fig. 2b). By comparing the near-field spectrum with the momentum- and frequency-dependent Fresnel reflection coefficient $r_p(q,\omega)$ for $p$-polarized radiation (Fig. 3b), it is

straightforward to associate them with the two symmetric ENZ modes in SrTiO$_3$ starting at $\omega_{LO,2}$ and $\omega_{LO,3}$ at the light cone ($q = \omega/c$) and showing a weak negative dispersion as the momentum increases. They are therefore direct counterparts of the radiative Berreman modes inside the light cone observed in off-normal incidence reflection spectra (Fig. 2b).

Remarkably, there are no features related to the transversal mode at $\omega_{TO,3}$, neither in the measured spectrum nor in the calculated dispersion for the gold-supported membrane. To establish whether this absence is related to the metallicity of the substrate, we show in Fig. 3c the s-SNOM amplitude spectrum obtained on the part of the membrane that resides on SiO$_2$/Si. One can see that the spectrum in the case of an insulating substrate is drastically different. While the signatures of the ENZ modes are still present, several new structures appear, notably peaks at 490 and 560 cm$^{-1}$ as well as an upturn below 370 cm$^{-1}$, near the limit of the experimental range. The 490 cm$^{-1}$ peak (marked with '*') stems from the SiO$_2$ optical phonons as it is present in the spectrum of the bare SiO$_2$/Si substrate (Supplementary Note 5 and Supplementary Fig. 8). However, the 560 cm$^{-1}$ peak and the 370 cm$^{-1}$ upturn do not match any phonons in SiO$_2$. To understand their provenance, we present in Fig. 3d the corresponding dispersion map for the SrTiO$_3$/SiO$_2$/Si structure. Now, two antisymmetric strongly dispersing modes of SrTiO$_3$ with a positive group velocity $d\omega/dq$ and a non-dispersing mode of SiO$_2$ (marked with '*') are present, in addition to the ENZ modes with a weak negative dispersion already seen in Fig. 3b. By a mere frequency comparison, the 560 cm$^{-1}$ peak and the 370 cm$^{-1}$ upturn can now be related to the antisymmetric modes.

Yet, comparing the s-SNOM spectra and the dispersion maps is not entirely straightforward, since the AFM tip interacts with the PhPs with a broad range of momenta. Therefore, we also simulate the spectra using the extended-dipole model (black lines in Fig. 3a, c)[39,40] based on the dielectric function of bulk SrTiO$_3$ (Fig. 2a). The model curves reproduce all the experiment features remarkably well. The fact that the match is quantitatively not perfect is not surprising given the approximative character of the substitution of a real tip with an ellipsoidal extended dipole in this approach. We can conclude that the s-SNOM amplitude increases in the spectral band, where the antisymmetric mode is seen in the dispersion. The attribution of the 560 cm$^{-1}$ peak and the 370 cm$^{-1}$ upturn to the antisymmetric modes is thus confirmed.

Why is the antisymmetric mode only seen in the SiO$_2$/Si-supported membrane, while the symmetric (ENZ) mode is observed on both substrates? To understand this, we perform finite-difference time-domain (FDTD) simulation. and Fig. 3f–i present the simulated distribution of the electromagnetic field in the SrTiO$_3$ membrane on both substrates at two frequencies (787 cm$^{-1}$ and 560 cm$^{-1}$) corresponding to the excitation of modes with different symmetry. Here, a plane wave is incident on the sample, and a small piece of gold is placed on top of the membrane to provide the necessary in-plane momentum. Figure 3f, h shows the field distribution for the case of the ENZ mode in SrTiO$_3$ on gold and SiO$_2$/Si, respectively. In accordance with the known properties of the ENZ modes[2,3], the field is concentrated inside the membrane, with a very weak intensity in the substrate. Thus, the substrate has only a limited influence on the ENZ-PhPs. Figure 3g, i illustrates the field distribution in the membrane on two different substrates when the antisymmetric SPhPs are excited. In contrast to symmetric ENZ-PhPs, the strongest electric field intensity is observed outside the membrane. Obviously, the antisymmetric modes are fully suppressed in the gold-supported membrane since the high conductivity of the metal does not allow the field to spread outside SrTiO$_3$.

## Propagation of SPhPs
Unlike ENZ modes, the antisymmetric SPhPs, having a positive group velocity, are expected to propagate and cause interference effects when

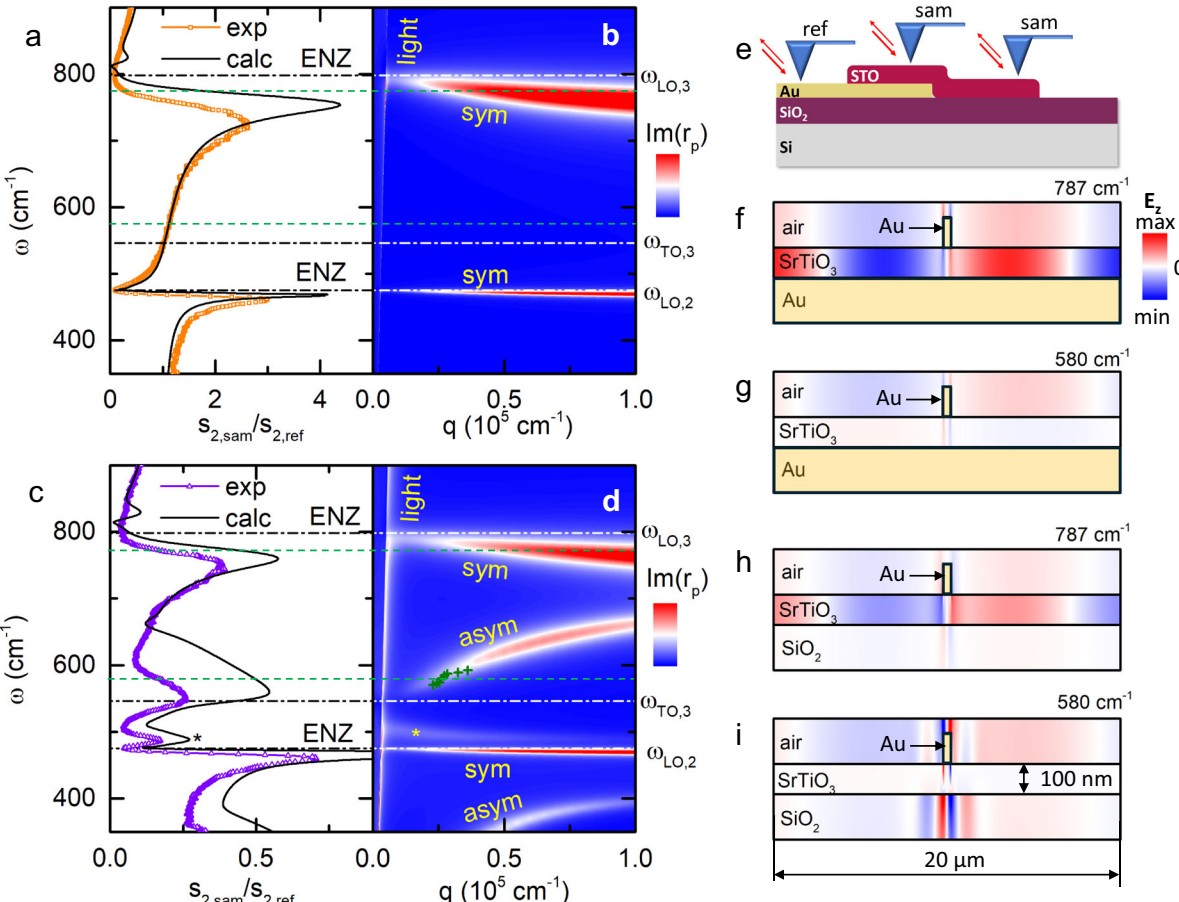

**Fig. 3 | SINS spectra on a SrTiO₃ membrane supported by gold- and SiO₂.**
**a**, **c** Symbols: SINS second-harmonics amplitude spectra ("sam") of the membrane on gold- (**a**) and on SiO₂/Si (**c**) normalized to the signal from bare gold ("ref"). Solid lines: finite-dipole model based on the dielectric function of bulk SrTiO₃.
**b,d** Calculated imaginary part of the complex reflection coefficient $r_p(q,\omega)$ for the 100 nm SrTiO₃ membrane on gold and on SiO₂/Si, respectively. Dashed-dotted Vertical lines in (**a**–**d**) denote the LO and TO frequencies of bulk SrTiO₃. "sym" and "asym" refer to symmetric and antisymmetric modes, respectively. Green crosses in

Fig.3d represent the data points in Fig.4g. **e** Schematic describing the SINS spectra acquisition. **f**–**i** Simulated distribution of the electromagnetic field, when the sample is illuminated with a plane wave and a gold nanobeam (marked with a black rectangle) is placed on top in order to excite waves with finite in-plane momenta, for the membrane on gold at 787 and 560 cm⁻¹, respectively (**e**, **f**), and for the membrane on SiO₂/Si at the same frequencies (marked with dashed lines in (**a**–**d**)). **h**, **i** In gold, the electromagnetic field is negligibly small.

the tip is near the sample edges. To verify this, we perform SINS line scans by moving the tip across the edge of the membrane on the SiO₂/Si substrate and recording the near-field spectra (such as the ones on Fig. 3a, c) as a function of $x$. Figure 4a, b shows the position-frequency maps of the s-SNOM amplitude and phase respectively, while Fig. 4c, d present the corresponding spectra at selected positions. As expected, only the spectral band between about 550 and 600 cm⁻¹ corresponding to the antisymmetric mode reveals a significant position dependence of both amplitude and phase (we exclude the area within about 200 nm from the edge, where topography-related artifacts may influence the s-SNOM spectra). Dashed red lines in Fig. 4a–d emphasize the spatially dispersing features, which we argue to be signatures of the first inter-ference fringe caused by the PhP propagation, akin to what has been observed in SNOM imaging of short-range plasmons in Bi₂Se₃[41] and phonon polaritons in atomically thin hBN[42].

To corroborate this argument, in Fig. 4e, f we plot the spatial profiles of the near-field amplitude, $s_2(x)$, and phase, $\phi_2(x)$, at three selected frequencies inside the 550–600 cm⁻¹ band. All profiles can be well fitted using the complex values parametrization: $s_2(x)e^{i\phi_2(x)} = A\frac{e^{iqx}}{x} + B$ as shown by solid red lines. Here $A$ and $B$ are complex fitting parameters, and $q$ is the complex wavevector $q = q_1 + iq_2$[36]. The first term in this equation represents the propagating wave launched by the edge and scattered by the tip to free space or

inversely launched by the tip and scattered by the edge, the second term is the s-SNOM signal far from the edge. As shown in Supplementary Fig. 9, this gives rise to a spiral-like dependence between the real and the imaginary parts of the near-field signal reminiscent of the plasmonic behavior observed earlier[41]. Notably, we do not need to introduce a term proportional to $e^{2iqx}$ corresponding to polaritons launched by the tip, reflected from the edge and scattered back to free space by the tip (Supplementary Note 6 and Supplementary Fig. 10). A likely reason is that in this process the PhPs travel a double distance and are therefore stronger damped as compared to the single passage.

In Fig. 4g, the real part of the extracted momentum, $q_1$, is shown as a function of frequency (symbols) and the same data points are pre-sented as crosses in Fig. 3d. The observed momenta are in excellent agreement with theory (solid line) and much higher than the ones in bulk SrTiO₃ (dashed line). The confinement factor $C = q_1/k_0$, where $k_0 = \omega/c$, is shown in Fig. 4h for the 100 nm membrane (diamonds) and bulk crystals (circles). In the first case, C is reaching 10 and it is, as expected, close to 1 in the latter case, indicating little confinement. Figure 4i shows the propagation length $L_p = 1/q_2$ and the quality factor $Q = q_1/q_2$. The value of $L_p$ ranges from 6 µm to 2 µm, decreasing with frequencies At the same time, the quality factor varies between 2 and 6, which is comparable to the value measured for the infrared plasmons in non-encapsulated graphene ($Q = 5$)[37,43] and larger than the value

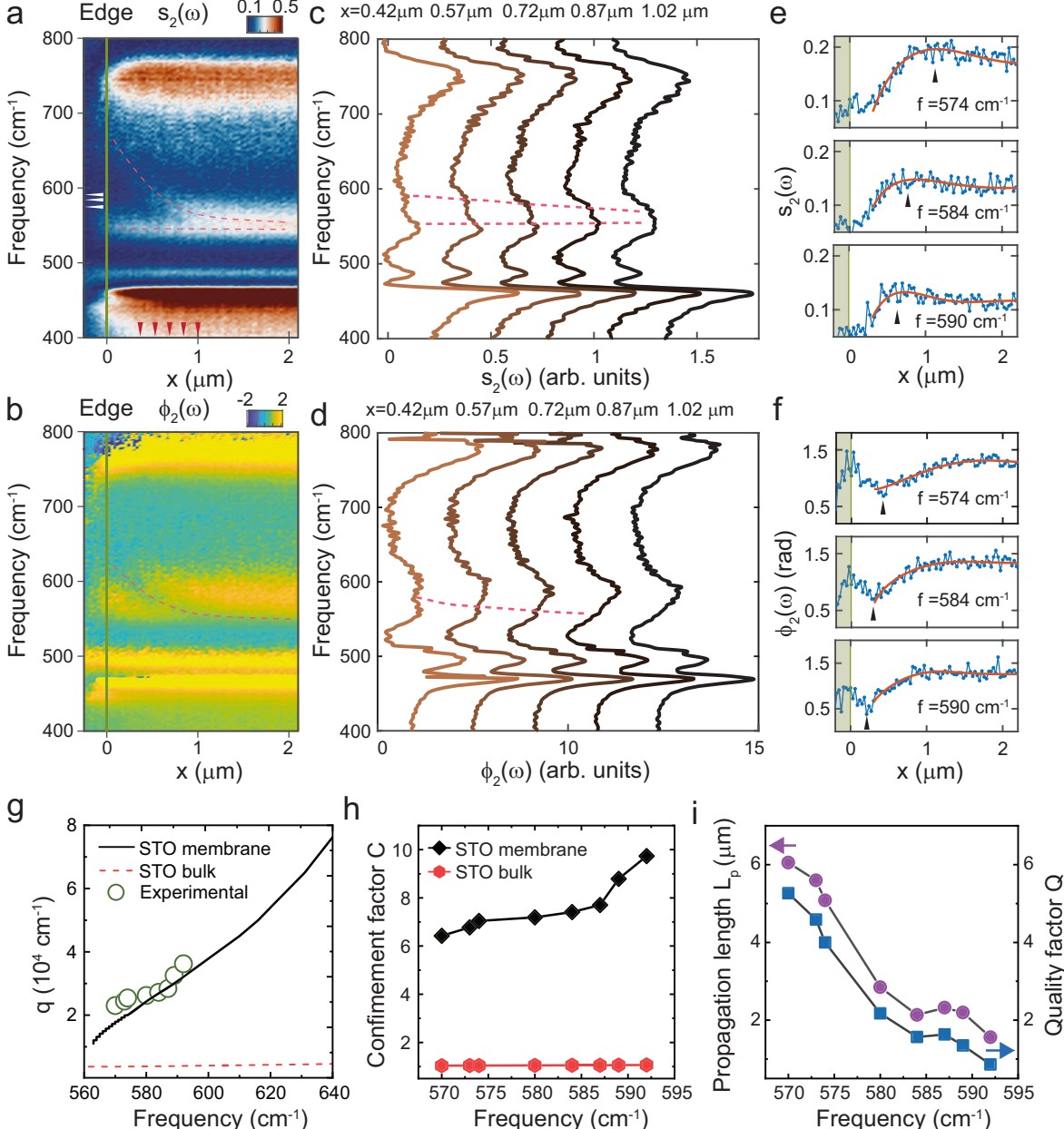

**Fig. 4 | Real-space SINS nanoimaging of SPhPs in SiO₂/Si-supported membranes. a**, **b** Experimental near-field amplitude (**a**) and phase (**b**) spectra obtained by a line scan perpendicular to the edge of the membrane. The trace of SINS linescan is shown in Fig. 1c. Dashed curves in (**a**) and (**b**) denote the peak and dip associated with the propagation of SPhSs. **c**, **d** Near-field SINS amplitude (**c**) and phase (**d**) spectra obtained at locations on the membrane with different distances to the edge. The locations are denoted by the red arrows in (**a**). Dashed curves in (**c**) and (**d**) denote the peak and dip associated with the propagation of SPhSs. **e**, **f** Simultaneously measured near-field amplitude (**e**) and phase (**f**) line profiles of the SrTiO₃ film at different frequencies (denoted by white arrows in (**a**)), with the scan direction perpendicular to the edge (denoted by green line) at $x = 0$. Experimental data is shown in blue. Black arrows in (**e**) and (**f**) show the shift of the peak in the amplitude profile and shift of the dip in the phase profiles. Red solid lines show the fitting of the experimental data using the complex-valued function $s_2(x)e^{i\phi_2(x)} = A\frac{e^{iqx}}{x} + B$. **g** Dispersion of SPhPs in bulk SrTiO₃ and membranes. Experimental data extracted from SINS imaging is shown as green circles. **h** The confinement factor of SPhPs in membranes and bulk SrTiO₃ as a function of frequency. **i** Propagation length $L_p$ (purple circles) and quality factor Q (blue squares) of SPhPs in SrTiO₃ membrane versus frequency.

measured for the terahertz plasmon in Bi₂Se₃ thin films ($Q = 3.2$)[41]. However, it is significantly smaller than the quality factor for plasmons in hBN encapsulated graphene (~25)[36], hyperbolic PhPs in hBN (~20)[44] and MoO₃ (~20)[9], as well as weakly confined SPhPs in bulk SrTiO₃ (~25)[14].

We note that no spatially dispersing structures are found in the similar SINS scans across the edge where the membrane is supported by gold (Supplementary Fig. 11). This is fully in agreement with the mentioned above absence of the antisymmetric mode found in this case, corroborating our conclusion that propagating antisymmetric SPhP modes are suppressed in metal-supported membranes.

## Discussion

In summary, we have experimentally observed a symmetric-antisymmetric phonon-polariton mode splitting in a high-quality single crystalline membrane of SrTiO₃. Due to the ultra-subwavelength sample thickness, the symmetric mode is pushed to the epsilon-near-

zero regime, where the normal component of the electric field is multiple times enhanced inside the sample as compared to the external field. The propagating antisymmetric mode is found to be confined in terms of the wavelength by a factor of 10 as compared to the SPhP mode in bulk SrTiO$_3$. The strong confinement of light and the field enhancement observed here can be exploited for various nanophotonic applications such as mid-infrared subwavelength resonators and metasurfaces. Transferable membranes of SrTiO$_3$ and other oxides are therefore a promising platform to realize perfect absorption and nonlinear nanophotonic applications in the infrared regime. In addition, enhanced light–matter interactions can enable the control over the paraelectric-to-ferroelectric phase transition in this material[45,46]. Notably, such membranes can be easily integrated with other photonic materials and resonators to form strongly coupled hybrid polaritons for light manipulation. Although we observed an extrinsic optical loss introduced by the SiO$_2$/Si substrate, the SPhPs propagation in the membranes can be potentially enhanced by using less lossy substrates or even using suspended membranes[47,48]. Overall, our work demonstrates the large potential of transition-metal oxide membranes as building blocks for future long-wavelength nanophotonics.

## Methods

### Thin-film growth
The epitaxial heterostructure of 100 nm SrTiO$_3$ films was synthesized with a 16 nm Sr$_2$CaAl$_2$O$_6$ sacrificial layer on (001)-oriented SrTiO$_3$ substrates via pulsed-laser deposition. The growth of the Sr$_2$CaAl$_2$O$_6$ layer was carried out in dynamic argon pressure of $4 \times 10^{-6}$ Torr, at a growth temperature of 710 °C, a laser fluence of 1.35 J/cm$^2$, and a repetition rate of 1 Hz, using a 4.8 mm$^2$ imaged laser spot. The growth of the SrTiO$_3$ layer was conducted in dynamic oxygen pressure of $4 \times 10^{-6}$ Torr, at a growth temperature of 710 °C, a laser fluence of 0.9 J/cm$^2$, and a repetition rate of 3 Hz, using a 3.0 mm$^2$ imaged laser spot.

### SrTiO$_3$ membrane fabrication
A 600 nm thick polymethyl methacrylate (PMMA) support layer was first spin coated on top of the heterostructure and baked at 135 °C. The heterostructure was then placed in deionized water at room temperature until the Sr$_2$CaAl$_2$O$_6$ had been fully dissolved. Prior to the membrane transfer, a layer of 50 nm Au was coated on one-half of a SiO$_2$/Si substrate by electron beam evaporation, such that half of the substrate was covered with Au whereas the other half was bare SiO$_2$/Si. The PMMA-coated SrTiO$_3$ film was then released from the SrTiO$_3$ substrate and transferred onto the prepared substrate. The PMMA support layer was then removed by dissolving in acetone at 60 °C and then washing in isopropanol, leaving just the SrTiO$_3$ membrane.

### Materials characterization
The AFM measurements were taken in tapping mode with a Veeco Multimode IV AFM equipped with a SPECS Nanonis 4 controller using MikroMasch HQ:NSC15/Al BS AFM tips with a force constant of ~40 Nm$^{-1}$. The XRD symmetric $\theta$-$2\theta$ line scans were performed using a Bruker D8 Discover with a monochromated Cu $K_{\alpha 1}$ ($\lambda = 1.5406$ Å) source. The atomic-resolution STEM imaging was performed using a probe aberration-corrected ThermoFisher Titan scanning transmission electron microscope operating at 300 keV.

### Far-field infrared spectroscopy
The far-field reflectivity measurements were performed using an infrared microscope Bruker Hyperion 2000 attached to a Fourier-transform infrared (FTIR) spectrometer Bruker Vertex 70 V. Two reflective focusing objectives were used having the numerical apertures (NA) of 0.4 and 0.5 (amplification of 15 and 36 times respectively). In the mid-infrared range, an MCT detector was used while in the far-infrared range a liquid He-cooled Si bolometer was utilized.

### Synchrotron infrared nanospectroscopy (SINS)
SINS experiments were performed at the Advanced Light Source (ALS) Beamline 2.4. The beamlines use an optical setup comprising of an asymmetric Michelson interferometer mounted into a commercial s-SNOM microscope (NeaSnom, Neaspec GmbH), which can be basically described by an AFM microscope possessing a suited optical arrangement to acquire the optical near-field. In the interferometer, the incident synchrotron IR beam is split into two components by a beamsplitter defining the two interferometer arms formed by a metallic AFM tip and an IR high-reflectivity mirror mounted onto a translation stage. The IR beam component of the tip arm is focused by a parabolic mirror on the tip–sample region. In the experiment, the AFM operates in semi-contact (tapping) mode, wherein the tip is electronically driven to oscillate (tapping amplitude of ~100 nm) in its fundamental mechanical frequency $\Omega$ (~250 kHz) in close proximity to the sample surface. The incident light induces an optical polarization to the tip, primarily, caused by charged separation in the metallic coating, the so-called antenna effect. The optically polarized tip interacting with the sample creates a local effective polarization. The back-scattered light stemming from this tip–sample interaction, is combined on the beamsplitter with the IR reference beam from the scanning arm and detected with a high-speed IR detector. A lock-in amplifier having $\Omega$ as the reference frequency demodulates the signal and removes the far-field contributions. The resulting interference signal is Fourier-transformed to give the amplitude $s_n(\omega)$ and phase $\phi_n(\omega)$ spectra of the complex optical $S_n(\omega)=s_n(\omega)e^{i\phi n(\omega)}$. All SINS spectra were measured for $n = 2$, i.e., $s_2(\omega)$ and $\phi_2(\omega)$. A customized Ge:Cu photoconductor, which provides broadband spectral detection down to 350 cm$^{-1}$, and a KRS-5 beamsplitter were employed for the far-IR measurements. The spectral resolution was set as 10 cm$^{-1}$ for a Fourier processing with a zero-filling factor of 4. All spectra in this work were normalized by a reference spectrum acquired on a clean gold surface.

**Calculation of the Fresnel reflection coefficient.** In order to model the far-field and near-field (SNOM) spectra for a multilayer sample, it is required to know the momentum- and frequency-dependent complex reflection coefficient $r_p(q,\omega)$ (p-stands for the p-polarization, since only the p-polarized light is considered). It can be computed using standard techniques, such as a recursive method described in the supplementary information of ref. 24 The thickness of the SrTiO$_3$ membrane (100 nm) is known from the AFM measurements and fixed in all simulations. The thickness of SiO$_2$ (275 nm) was deduced from the position of the reflectance interference minimum in the mid-infrared range (about 6000 cm$^{-1}$) since the dielectric function of SiO$_2$ above the phonon range is well known. The gold layer and the silicon layer were considered as infinitely thick as both materials for the actual thickness (50 nm of Au and 1 mm of doped Si) are opaque to radiation in the frequency range of interest.

**Finite-dipole simulations.** We use the standard finite-dipole model (FDM), where the tip is represented with an ellipsoid[39,40]. The same FDM parameters were used in all simulations: the tapping amplitude 76 nm (peak-to-peak), the lowest tip position with respect to the sample $b = 0$, the angle of incidence $\alpha = 45°$, the half-length of the main axis of the ellipsoid $L = 740$ nm and the tip radius $a = 100$ nm.

**FDTD simulations.** Field distribution of surface phonon polaritons and ENZ modes is simulated using Lumerical FDTD software. Optical dielectric functions of SrTiO$_3$, SiO$_2$, and Si extracted from our far-field reflection measurements are used for the simulation. Normally, the incident plane waves are used as the excitation source, with an infinitely long gold nanobeam measuring 100 nm in height and 400 nm in width positioned atop the SrTiO$_3$ membrane. This arrangement is designed to impart in-plane momentum for exciting surface polaritons.

## Data availability

All data that support the findings of this study are present in the paper and the Supplementary Information. Further information can be obtained from the corresponding author upon request.

## Code availability

All code used in this study is available from the corresponding author upon request.

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

## Acknowledgements

This material is based upon work supported by the National Science Foundation under Award No. DMR-2340751. R. X. acknowledges the support from the Faculty Research and Professional Development Program (FRPD) at North Carolina State University. This material is based upon work supported by the National Science Foundation, as part of the Center for Dielectrics and Piezoelectrics under Grant Nos. IIP-1841453 and IIP-1841466.Work at Stanford/SLAC was supported by the Department of Energy, Office of Basic Energy Sciences, Division of Materials Sciences and Engineering, under contract no. DE-AC02-76SF00515. Partial support (for J.A.D. and Y.L) was also provided by the U.S. Department of Energy Office of Science National Quantum Information Science Research Centers. This research used resources of the Advanced Light Source, a U.S. DOE Office of Science User Facility under contract no. DE-AC02-05CH11231. The authors thank Christopher Winkler for assistance with the TEM imaging at North Carolina State University. This work was performed in part at the Analytical Instrumentation Facility (AIF) at North Carolina State University, which is supported by the State of North Carolina and the National Science Foundation (award number ECCS-2025064). The research of I.C., Y.Z.,A.B., L.K., C.W.R., J.T., and A.B.K. is supported by the Swiss National Science Foundation (grant #200020_201096).

## Author contributions

R.X. and I.C. contributed equally to this paper. Y.L. and A.B.K. conceived the study with the help of J.A.D. R.X., K.J.C, Y.L., J. Li, and H.Y.H. synthesized and characterized the membranes. R.X., H.A.B., S.N.G., Y.Z. and A.B. performed the s-SNOM measurements. I.C., A.B.K., C.W.R., and J.T. performed the far-field measurements. Y.L., I.C. and A.B.K performed the simulation and data analysis. L.K. performed the AFM measurements. R.X., I.C., A.B.K., and Y.L. wrote the manuscript with input from all the authors. All authors contributed to scientific discussion.

## Competing interests

The authors declare no competing interests.
