## [Peer Review File · Nature Communications]

Highly confined epsilon-near-zero and surface-phonon polaritons in SrTiO₃ membranesREVIEWER COMMENTS

Reviewer #1 (Remarks to the Author):

Highly confined epsilon-near-zero- and surface-phonon polaritons in SrTiO₃ membranes
Xu et al.

In this work, the authors investigate surface phonon polaritons in SrTiO₃ membranes. They use s-SNOM and FTIR to reveal the confinement factor and loss factor of ENZ and surface polaritons. Extensive modeling is presented.

Despite a few awkward sentences, the manuscript is well written, structured, and easy to read. The experiments described are interesting, as is the sample structure investigated. The instrumentation used stands out. The quality of the data is good, the analysis is rigorous and the claims are generally well supported. The modeling significantly add to the quality of the manuscript.

Surface phonon polaritons have been studied in a variety of materials and thin layers, and large confinement factors have already been demonstrated. ENZ-modes were very well discussed in papers by Vassant et al. (For instance, PRL 109, 23701 (2012)). The novelty of this work appears to lie in the SrTiO₃ membranes. Is this enough to bring the novelty to the level required? I am not so sure. In recent years, guided and surface hyperbolic polaritons in anisotropic materials have been in the spotlight. The literature on Van der Waals crystals is all too briefly mentioned, and the performance metric of SrTiO₃ do not stand out. Hence, despite the quality of the investigation, I am not of the impression that this manuscript meet the criterion for publication in this journal.

Having (carefully) read the manuscript, I would like to transmit the following comment to the authors.

Comments:

Everywhere: The authors discuss polaritons in terms of ENZ polaritons and surface polaritons. This choice is confusing, since the ENZ-PhP is the antisymmetric surface polariton that was pushed to the LO frequency (the ENZ range) in the thin sample limit. If the authors truly believe that they are fundamentally different, this aspect should be better discussed.

Abstract and elsewhere: The use of "hyperspectral" is not clear and should be clarified. My understanding is that "hyperspectral techniques" speed up the 3D acquisition (x,y, lambda) using at least one form of 2D acquisition. The SINS experiment is not hyperspectral since the sample and the interferometer must sequentially scanned. Is the FTIR reflection measurement performed with an imaging FTIR (2D sample image + 1D interferometer scanning)?

Abstract and text : The authors claim highly confined and high momentum SPhPs, isn't these two sides of the same phenomenon?

Page 3: Only the limitations of Van der Waals compounds are presented. However, these materials have some advantages that SrTiO₃ cannot provide, such as volume phonon-polaritons. A more balanced assessment should be presented.

Page 4: "in the frequency window where the epsilon is zero." There is no such thing.

Page 5: The authors should add a few sentences on the relevance of performing two studies, one on Au and one on SiO₂/Si. As it is, it does not stand out. Why is it important to present and discuss the Berreman mode?

Page 5: "with a sharp and straight edge". From Fig 1c and 1d, the edge does not appear particularly sharp.

Page 5 and Supp. Data: Uncertainties on fitting parameters should be provided.

Abstract and elsewhere in the text: "far-field spectroscopy", "infrared microscope", "near-field nanoimaging" are very vague descriptions, it would be better to be more specific (FTIR, s-SNOM).

Fig. 3. How can Fig. 3 (b-d) be related to the calculated curve in Fig. 3(a-c)? One is q-resolved and the other is not. What does it say about the q-selectivity of the s-SNOM measurement?

Page 7: How is the red shift of wTO,3 determined? A fit? This should be shown in S.I. along with the resulting uncertainties.

Page 8: S₂: The meaning of the 2 should be mentioned right away.

Page 9: Why is the field simulation done at 593 cm⁻¹? There is little experimental signal at 593. Would 560 cm⁻¹ not be better? Please clarify. Also, why 787 cm⁻¹ ?

Page 9: "SPhPs are fully suppressed...". Page:10: "SPhPs, unlike ENZ modes". Please clarify in the context of the ENZ being a surface polariton.

Page 12: γ^{-1} is compared with values for plasmons. What about a comparison with values for other phPols? With phPols in VDW crystals for instance.

Reviewer #2 (Remarks to the Author):

The authors observed epsilon-near-zero (ENZ) and surface phonon polariton (SPhP) modes in the thin SrTiO₃ (STO) membrane by incorporating near-field excitation/detection methods. This paper is well-written and contains some interesting materials. In particular, the recent advances on the STO membranes implicate significant potential in various interesting research areas; this work would be the first to observe the ENZ and SPhP modes in these platforms. In contrast, the present manuscript is rather premature and does not contain significant advances in optical science or technology. Besides, their claim is not fully supported by their experimental results in the current manuscript. Therefore, I would not recommend publication in Nature Communications. My concerns/questions are listed below:

- (1) Importantly, they need to provide more compelling arguments regarding the unique characteristics in the STO membrane, compared to those in atomically thin TMD materials and other (oxide) films. Large-area growth and tunability are not exclusive features of the STO membrane. An example in the mid-IR range (taken with a FT-IR microscope) can be found in the literature (Park et al., 10.1038/srep15754).
- (2) The authors need experimental verification for Fig. 3b&d (at least a few points), which is one of the key elements to support their claims.
- (3) Considering the long wavelength, subwavelength patterning on the membrane would be beneficial, potentially enabling far-field excitation.
- (4) Why do they need the transferred STO membrane in the first place? For instance, a thin STO film grown on Si and SiO₂ has been available.
- (5) I'm wondering if they considered SiO₂ having a natural ENZ mode at 8 μm .
- (6) Please do not use the term "terahertz" in the abstract/summary as it can be misleading. The spectral range of interest is Mid-IR, rather than the Far-IR.
- (7) (technical error) Fig. S6 (and inset) does not have blue symbols (solid lines instead).

Reviewer #3 (Remarks to the Author):

It is my pleasure to serve as a referee for the manuscript submitted by R. Xu et al. I hope that the comments provided herein will serve to improve the work.

In the present effort, the authors describe the preparation of a sample containing a thin film of crystalline SrTiO₃ transferred to a SiO₂ substrate partially coated with gold. Using this sample the authors measure far field and near field responses of various phononic modes within the sample in the range of 400 – 800 cm⁻¹. The authors compare and contrast the presence of and features exhibited between modes observed in the SrTiO₃-Au and SrTiO₃-SiO₂ portion of the film. In particular, the note the presence of ENZ modes (both cases) surface phonon polariton modes (only SiO₂). Of note, the authors the confinement of the modes is high, ~10x more than in bulk films. Throughout the authors provide well-reasoned arguments, supported by simulation, to explain the dependencies they observe in experiment. Overall I found the paper to be technically sound, easy to read, and thorough. I have no technical concerns for the paper and believe the quality is worthy of publication in Nature Communications. The capability of the thin film SrTiO₃ to achieve high confinement combined with various substrate is likely to be of interest to a host of researchers in the area of MIR plasmonics.

General comment:

1) I found the discussion around figure 4a,b,g,h a little confusing and honestly could not capture what it was trying to tell me. I kept trying to make it look like a plot of propagation length like is done in SNOM, but I am not sure if this is the point. Perhaps the authors could include a physical image of the sample illustrating the path of the tip to compare with their measurements and clarify the discussion? Also perhaps rotating the panels c,d, so that the frequency axis matches with a,b and ordering the graphs from left to right in terms of increasing distance would help drive home the relationship between the two plots.

REVIEWER COMMENTS

Reviewer #1 (Remarks to the Author):

Highly confined epsilon-near-zero- and surface-phonon polaritons in SrTiO₃ membranes Xu et al.

In this work, the authors investigate surface phonon polaritons in SrTiO₃ membranes. They use s-SNOM and FTIR to reveal the confinement factor and loss factor of ENZ and surface polaritons. Extensive modeling is presented.

Despite a few awkward sentences, the manuscript is well written, structured, and easy to read. The experiments described are interesting, as is the sample structure investigated. The instrumentation used stands out. The quality of the data is good, the analysis is rigorous and the claims are generally well supported. The modeling significantly adds to the quality of the manuscript.

We are delighted to receive such a high appreciation.

Surface phonon polaritons have been studied in a variety of materials and thin layers, and large confinement factors have already been demonstrated. ENZ-modes were very well discussed in papers by Vassant et al. (For instance, PRL 109, 23701 (2012)). The novelty of this work appears to lie in the SrTiO₃ membranes. Is this enough to bring the novelty to the level required? I am not so sure.

In recent years, guided and surface hyperbolic polaritons in anisotropic materials have been in the spotlight. The literature on Van der Waals crystals is all too briefly mentioned, and the performance metric of SrTiO₃ does not stand out. Hence, despite the quality of the investigation, I am not of the impression that this manuscript meets the criterion for publication in this journal.

We thank the Reviewer for this fair comment. Indeed, ENZ-modes were extensively discussed in a number of papers, including the one which the reviewer mentions, and which has been cited in the original manuscript. The reviewer is right that the primary novelty of this work rests on using the SrTiO₃ membranes, which have never been used so far for phonon-polariton studies. Here we allow ourselves to elaborate on this. Complex and functional oxides are important electronic materials holding many useful intrinsic physical properties such as ferro/piezoelectricity, pyroelectricity, ferromagnetism, magnetostriction and superconductivity. These materials can also be complemented with conventional semiconductor-based devices or used on their own to realize electronic, spintronic and optical applications. However, integration of complex oxides onto conventional semiconductor platforms has been difficult up to now because of significant lattice mismatch with conventional semiconductor devices (such as Si or III-V/III-N materials). In the recent few years, methods to create freestanding single-crystalline, large-area complex oxide membranes have been developed, which allows heterogeneous integration of single-crystalline membranes with other materials to create novel electronic and photonic devices. Research in the oxide membranes has been growing rapidly and ferroelectric, magnetic, thermophysical, electronic properties of the membranes have been explored. Our work is the first experimental study on the mid-infrared polaritonic property of the membrane, which reveals highly confined ENZ and surface phonon-polaritons in the membrane, in contrast to bulk SrTiO₃.

We also agree with the Reviewer that the figures of merit of phonon polaritons in SrTiO₃ membranes currently do not outperform the ones in the record-holding vdW materials such as hBN and α -MoO₃. However, the ultrathin crystals of perovskite oxides bring other advantages. First, each polar material, including SrTiO₃, features a spectroscopically unique set of Reststrahlen bands thus adding to the palette of various applications. Second, there is a broad range of complex oxide materials including K₂TiO₃, LiNbO₃, BaTiO₃, and BiFeO₃ and heterostructures such as the SrTiO₃/LaAlO₃ hosting a 2D electron gas

that can be possibly made as transferable membranes, which could act as useful platforms for phonon polaritons. Our study is the first demonstration of the enormous potential of this large family of complex oxide materials. Third, all vdW materials are optically anisotropic, and this anisotropy in some cases may be a significant obstacle for applications.

To address this remark of the Referee, in the revised manuscript we explicitly mention that the PhP figures of merits in the studied membrane are inferior to hBN and encapsulated graphene. We also significantly rewrote the introduction and conclusions, while including more references, in order to emphasize the points made above, which justify interest in the oxide membranes, especially in the context of polaritonics.

Having (carefully) read the manuscript, I would like to transmit the following comment to the authors.

Comments:

Everywhere: The authors discuss polaritons in terms of ENZ polaritons and surface polaritons. This choice is confusing, since the ENZ-PhP is the antisymmetric surface polariton that was pushed to the LO frequency (the ENZ range) in the thin sample limit. If the authors truly believe that they are fundamentally different, this aspect should be better discussed.

This is a good point and we admit the terminology was ambiguous in the original version. Indeed, both the symmetric and antisymmetric modes originate from the same SPhP in bulk crystals, as the thickness decreases. As a matter of fact, the various notations in the literature on this subject are often mutually inconsistent. In the table and the figure below, we make a review of different articles demonstrating this diversity, which potentially creates confusion for the readers.

our work	SrTiO₃	Berreman mode	antisymmetric mode	symmetric mode /ENZ mode
Kinsey, N., DeVault, C., Boltasseva, A. & Shalaev, V. M. Near-zero-index materials for photonics. Nature Reviews Materials 4 , 742–760 (2019).	review	Berreman mode Brewster mode Leaky mode	Short-range mode	Long-range mode
N.K. Passler et al. ACS Photonics 2019, 6 , 1365–1371. Also Nano Letters 2018, 18 , 4285	AlN/SiC, PhP	Berreman mode radiative	Thin film polariton Antisymmetric evanescent (SPhP term reserved for thick samples)	ENZ thin film polariton Symmetric Evanescent ENZ(AlN)-SPhP (SiC)

Mancini, A., ... Mayer et al. Near-field retrieval of the surface phonon polariton dispersion in free-standing silicon carbide thin films. ACS Photonics 9, 3696–3704 (2022).	SiC, PhP		Odd mode	Even mode ENZ mode Weakly propagating mode
S. Vassant et al, Epsilon-Near-Zero Mode for Active Optoelectronic Devices. PRL 109, 237401 (2012)	GaAs/AlGaAs, PhP		Antisymmetric mode	ENZ-SPhP mode ENZ mode Symmetric mode
Campione, S., Brener, I. & Marquier, F. Theory of epsilon-near-zero modes in ultrathin films. Phys. Rev. B Condens. Matter 91, 121408 (2015).	Theory, mentioning SPP, But of general application	Berreman mode Brewster mode	Short-range SPP	ENZ mode Long-range SPP
D. N. Basov,* M. M. Fogler, F. J. García de Abajo, Polaritons in van der Waals materials, Science 354, aag1992 (2016).	review		Symmetric	Asymmetric
D. Sarid, PRL 47, 1927 (1981)	Thin metal films		Antisymmetric Evanescent Short range	Symmetric Evanescent Long range
S. Vassant et al, Berreman mode and epsilon near zero mode, Optics Express 20, 23971, 2012	Theory, applied to QWs	Berreman leaky mode	Surface phonon polariton mode	ENZ mode

Kinsey, N., DeVault, C., Boltasseva, A. & Shalaev, V. M. Near-zero-index materials for photonics. *Nature Reviews Materials* 4, 742–760 (2019).

S. Vassant et al. Epsilon-Near-Zero Mode for Active Optoelectronic Devices. *PRL* 109, 237401 (2012)

N.K. Passler et al. *ACS Photonics* 2019, 6, 1365–1371

S. Vassant et al. Berreman mode and epsilon near-zero mode *Optics Express* 20, 23971, 2012

Mancini, A. et al. Near-field retrieval of the surface phonon polariton dispersion in free-standing silicon carbide thin films. *ACS Photonics* 9, 3696–3704 (2022).

Campione, S., Brenner, I. & Marquier, F. Theory of epsilon-near-zero modes in ultrathin films. *Phys. Rev. B Condens. Matter* 91, 121408 (2015).

FIG. 3. The absolute value of the transverse magnetic field distribution of the (a) long-range and (b) short-range modes (not to scale).

D. Sarid, *PRL* 47, 1927 (1981) (> 700 citations !)

D. N. Basov,* M. M. Fogler, F. J. Garcia de Abajo, Polaritons in van der Waals materials. *Science* 354, aag1992 (2016).

To make things as consistent as possible with the literature, we choose the following terminology. The terms “symmetric” and “antisymmetric” refer to respectively even and odd symmetry z-axis dependence of the z-axis component of the electric field (z is normal to the sample) with respect to the center of the membrane. We note that if the in-plane component of the field were chosen as a symmetry criterion then the terminology would flip.

Within this notation, the ENZ mode is a symmetric mode, when it is pushed to the LO frequency, where the permittivity (epsilon is zero. The Berreman mode is the low-momentum extension of the ENZ mode inside the light cone, where it becomes leaky (radiative).

In the revised version, we carefully introduce this notation, and consistently use it throughout the paper.

Abstract and elsewhere: The use of “hyperspectral” is not clear and should be clarified. My understanding is that “hyperspectral techniques” speed up the 3D acquisition (x,y, lambda) using at least one form of 2D acquisition. The SINS experiment is not hyperspectral since the sample and the interferometer must sequentially scanned. Is the FTIR reflection measurement performed with an imaging FTIR (2D sample image + 1D interferometer scanning)?

We used the term “hyperspectral imaging” to denote a measurement, where s-SNOM spectra are obtained sequentially, while the tip is positioned at different spatial points, typically along a 1D line (it can be a 2D scan as well, which is much more time consuming though). This term is used by the neaspec company (<https://www.attocube.com/en/products/microscopes/nanoscale-imaging-spectroscopy/technology/nano-FTIR>) and reproduced in many papers. There, the meaning of “hyperspectral” is that multiple wavelengths are probed at the same time. However, it was easy for us to remove this word everywhere in the text and instead explain what is actually measured.

Abstract and text : The authors claim highly confined and high momentum SPhPs, isn't these two sides of the same phenomenon?

Indeed, this is exactly the same thing. We modified the abstract and the text accordingly.

Page 3: Only the limitations of Van der Waals compounds are presented. However, these materials have some advantages that SrTO₃ cannot provide, such as volume phonon-polaritons. A more balanced assessment should be presented.

We thank the reviewer for the comment. As mentioned above, in the revised version we provide a balanced assessment of the advantages of both vdW materials and the complex oxides and cite additional articles.

Page 4: “in the frequency window where the epsilon is zero.” There is no such thing.

Good point! This is fixed by replacing “is zero” with “close to zero”.

Page 5: The authors should add a few sentences on the relevance of performing two studies, one on Au and one on SiO₂/Si. As it is, it does not stand out.

We significantly corrected the logic flow, so that it becomes clear why it is important to compare gold- and SiO₂/Si supported membranes. Notably, the antisymmetric mode is fully suppressed on the metallic substrate, which is a very important observation for theory and applications. As a related point, the possibility to do these two measurements using an identical sample is a great advantage of using transferrable membranes. We believe this fact adds to the novelty and significance of our work.

Why is it important to present and discuss the Berreman mode?

As mentioned, the Berreman mode is a continuation of the ENZ dispersion branch to the radiative regime inside the light cone. The observations of the Berreman mode and the symmetric SPhP mode using far-field and near-field techniques respectively are therefore complementary and logically very relevant.

Page 5: “with a sharp and straight edge”. From Fig 1c and 1d, the edge does not appear particularly sharp.

We suspect that the Reviewer is confused because in the original version of Fig.1 it was less clear which edge is the one of gold and which is the one of the membrane. We modified the figure to make this unambiguous. One can see that the membrane edge (marked with the white arrow) is indeed very sharp. In the AFM, the sharpness of the high step (100 nm) is smeared by convolution with the tip shape.

Page 5 and Supp. Data: Uncertainties on fitting parameters should be provided.

In the revised version, we provide all fit uncertainties.

Abstract and elsewhere in the text: “far-field spectroscopy”, “infrared microscope”, “near-field nanoimaging” are very vague descriptions, it would be better to be more specific (FTIR, s-SNOM).

We thank the Referee for the suggestion. This is done.

Fig. 3. How can Fig. 3 (b-d) be related to the calculated curve in Fig. 3(a-c)? One is q-resolved and the other is not. What does it say about the q-selectivity of the s-SNOM measurement?

This is a very pertinent remark. In the s-SNOM experiment, modes with many q-momenta are excited simultaneously and indeed, the comparison between panels b,d and a,c is not entirely straightforward, even though they are intrinsically related. To help the reader make a connection, and also following a recommendation of the 3rd Reviewer, we swapped the axes in Fig.3a and 3c, in order to match the vertical (frequency) axis of the s-SNOM spectra and the dispersion maps. Now it is obvious that the calculated SPhP modes are related to the measured spectral features. Furthermore, the extended-dipole simulation (solid curves in Fig. 3a and 3c) make this connection more obvious. The discussion around Fig. 3 is significantly modified in the new version in order to address this comment.

Page 7: How is the red shift of $\omega_{TO,3}$ determined? A fit? This should be shown in S.I. along with the resulting uncertainties.

Yes, this was done via a fit, and we acknowledge that this was not well explained before. Following this recommendation, we added a new Supplementary note, where the fit is described in detail. The resulting parameter values are presented, together with their uncertainties. It is now well demonstrated that the shift is beyond the uncertainty.

Page 8: S_2 : The meaning of the 2 should be mentioned right away.

This is done.

Page 9: Why is the field simulation done at 593 cm^{-1} ? There is little experimental signal at 593. Would 560 cm^{-1} not be better? Please clarify. Also, why 787 cm^{-1} ?

Indeed, the frequency of 560 cm^{-1} is better! We now show the simulation at this frequency, although the result is qualitatively the same. The frequency of 787 cm^{-1} is chosen such that it is close enough to the LO frequency, but is still in the spectral region, where the s-SNOM signal is relatively high.

Page 9: “SPhPs are fully suppressed...”. Page:10: “SPhPs, unlike ENZ modes”. Please clarify in the context of the ENZ being a surface polariton.

This question is related to the discussed issue of terminology. In the revised manuscript, it is fully clarified.

Page 12: γ^{-1} is compared with values for plasmons. What about a comparison with values for other phPols? With phPols in VDW crystals for instance.

In the new version, we present this comparison and we fairly state that the PhP quality factor Q (which we now use instead of γ^{-1}) in the STO is inferior to that in hBN and MoO_3 .

Reviewer #2 (Remarks to the Author):

Editor:

The authors observed epsilon-near-zero (ENZ) and surface phonon polariton (SPhP) modes in the thin SrTiO_3 (STO) membrane by incorporating near-field excitation/detection methods. This paper is well-written and contains some interesting materials. In particular, the recent advances on the STO membranes implicate significant potential in various interesting research areas; this work would be the first to observe the ENZ and SPhP modes in these platforms.

We are grateful for this encouraging comment. Indeed, to the best of our knowledge we are the first to observe the symmetric-antisymmetric splitting in STO membranes and see so clearly the ENZ and Berreman physics in these systems.

In contrast, the present manuscript is rather premature and does not contain significant advances in optical science or technology. Besides, their claim is not fully supported by their experimental results in the current manuscript.

We agree that the original version of our manuscript did not emphasize well enough the novelty and the advances in optical science and technology. To address this remark and also to answer the other Reviewers, in the new version, we have significantly reworked the introduction and the discussion. Now we properly put our work in the context of modern research on other materials .

My concerns/questions are listed below:

(1) Importantly, they need to provide more compelling arguments regarding the unique characteristics in the STO membrane, compared to those in atomically thin TMD materials and other (oxide) films.

In the new version, we compare the achieved figures of merit on STO with other studies, notably on famous van der Waals systems, such as hBN, α -MoO₃ and graphene (see also our comments to other Reporters). To the best of our knowledge, the TMD materials did not show so far convincing mid- and far-infrared phonon-polaritonic performances (even though they demonstrate nice exciton polaritonic behavior in the near-infrared and the visible range, which is not the topic of this paper).

As we wrote in the reply to the 1st Referee, the figures of merit of phonon polaritons in SrTiO₃ membranes currently do not outperform the ones in the record-holding vdW materials such as hBN and α -MoO₃. However, the ultrathin crystals of perovskite oxides bring other advantages First, each polar material, including SrTiO₃ features a spectroscopically unique set of Reststrahlen bands thus adding to the palette of possible applications. Second, there is a broad range of complex oxide materials including KaTiO₃, LiNbO₃, BaTiO₃, and BiFeO₃ and heterostructures such as the SrTiO₃/LaAlO₃ hosting 2D electron gas that can be possibly made as transferable membranes, which could act as useful platforms for phonon polaritons. The results of our work potentially apply to these systems as well. Third, all vdW materials are optically anisotropic, and this anisotropy in some cases could be a significant obstacle for applications.

Large-area growth and tunability are not exclusive features of the STO membrane. An example in the mid-IR range (taken with a FT-IR microscope) can be found in the literature (Park et al., 10.1038/srep15754).

We thank the Referee for bringing this paper to our attention. This article, which we now cite in the main text, further demonstrates the general interest in the topic of ENZ materials. We note, however, that this paper exploits ITO with plasmon-polaritons, rather than phonon-polaritons as in our case. Furthermore, they report the phenomenon at wavelengths (3-5 micron) significantly shorter than ours (up to 21 micron).

(2) The authors need experimental verification for Fig. 3b&d (at least a few points), which is one of the key elements to support their claims.

Following this useful suggestion, we added experimental points (diagonal crosses) to Fig.3d. This makes indeed a close connection between Fig. 4 and Fig.3.

(3) Considering the long wavelength, subwavelength patterning on the membrane would be beneficial, potentially enabling far-field excitation.

We fully agree with the Reviewer that patterning is an obvious continuation of our work and indeed the large scale of single-crystalline STO membranes is a key advantage for such studies. However, such a development would be certainly beyond the scope of the current paper and would make it much longer and less readable. In the new version, we mention the possibility of patterning in the summary paragraph.

(4) Why do they need the transferred STO membrane in the first place? For instance, a thin STO film grown on Si and SiO₂ has been available.

In general, transferable membranes of complex oxides offer important advantages as compared to grown films. In particular, fabrication of electronic, spintronic and photonic devices often dictates integration of complex-oxides with conventional semiconductors, which is rather difficult in the case of grown films because of the lattice mismatch. For instance, ultrathin films of SrTiO₃ epitaxially grown on (001) Si show a biaxial compression due to a 1.7% mismatch between STO and silicon, while thicker films develop dislocations, which significantly reduce their quality, possibly including polaritonic quality factors (see for example, M.P. Warusawithana et al., *Science*, 324, 367 (2009) and Y. Tian et al., *Appl. Phys. Lett.* 102, 041906 (2013)). On the other hand, using freestanding single-crystalline, large-area complex oxide membranes allows heterogeneous integration of high-quality, strain-free, single-crystalline membranes with arbitrary materials to create novel electronic and photonic devices. These considerations, related to future applications, have determined our choice in this case.

(5) I'm wondering if they considered SiO₂ having a natural ENZ mode at 8 μm .

Yes! Indeed, the SiO₂ layer also shows a natural Berreman (ENZ) mode at 1260 cm⁻¹ (7.9 μm). This is shown in the Figure below. As we do not focus on SiO₂, we do not mention this in the text to avoid diverting the reader from the main logic flow.

(6) Please do not use the term “terahertz” in the abstract/summary as it can be misleading. The spectral range of interest is Mid-IR, rather than the Far-IR.

Following the reviewer’s suggestion, we removed entirely the term “terahertz” from the manuscript.

(7) (technical error) Fig. S6 (and inset) does not have blue symbols (solid lines instead).

It is fixed.

Reviewer #3 (Remarks to the Author):

It is my pleasure to serve as a referee for the manuscript submitted by R. Xu et al. I hope that the comments provided herein will serve to improve the work.

In the present effort, the authors describe the preparation of a sample containing a thin film of crystalline SrTiO₃ transferred to a SiO₂ substrate partially coated with gold. Using this sample the authors measure far field and near field responses of various phononic modes within the sample in the range of 400 – 800 cm⁻¹. The authors compare and contrast the presence of and features exhibited between modes observed in the SrTiO₃-Au and SrTiO₃-SiO₂ portion of the film. In particular, they note the presence of ENZ modes (both cases) surface phonon polariton modes (only SiO₂). Of note, the authors the confinement of the modes is high, ~10x more than in bulk films. Throughout the authors provide well-reasoned arguments, supported by simulation, to explain the dependencies they observe in experiment. Overall I found the paper to be technically sound, easy to read, and thorough. I have no technical concerns for the paper and believe the quality is worthy of publication in Nature Communications. The capability of the thin film SrTiO₃ to achieve high confinement combined with various substrate is likely to be of interest to a host of researchers in the area of MIR plasmonics.

We thank the Reviewer for the kind comment. And indeed, their comments helped us to improve the work significantly.

General comment:

1) I found the discussion around figure 4a,b,g,h a little confusing and honestly could not capture what it was trying to tell me. I kept trying to make it look like a plot of propagation length like is done in SNOM, but I am not sure if this is the point.

In the revised version, we significantly improved the Fig.4 and the discussion around it. We hope that it is now much clearer. In fact, by scanning the tip across the membrane edge, we are able to extract the important PhP parameters, such as the momentum, the confinement coefficient, the propagation length and the quality factor. We are also able to see which PhP bands contain propagating PhPs and which do not.

Perhaps the authors could include a physical image of the sample illustrating the path of the tip to compare with their measurements and clarify the discussion?

We thank the Referee for the suggestion. We added a line in Fig.1c indicating the path of the tip during this measurement.

Also perhaps rotating the panels c,d, so that the frequency axis matches with a,b and ordering the graphs from left to right in terms of increasing distance would help drive home the relationship between the two plots.

This is an excellent suggestion. We followed it both in Fig.4c, d as the Referee asked, and also in Fig.3a, c. Now the correspondence with the neighbor panels is straightforward. As an additional step to improve the figure clarity, we now use different color schemes for the phase and the amplitude frequency-position maps.

REVIEWERS' COMMENTS

Reviewer #1 (Remarks to the Author):

Highly confined epsilon-near-zero- and surface-phonon polaritons in SrTiO₃ membranes
Ruijuan Xu et al.

The authors have done a good job in clarifying or correcting a number of issues. I have one final comment. The authors wrote in their reply,

“To the best of our knowledge, the TMD materials did not show so far convincing mid- and far-infrared phonon-polaritonic performances (even though they demonstrate nice exciton polaritonic behavior in the near-infrared and the visible range, which is not the topic of this paper).”

I would like to bring to the attention of the authors this following work:

A. Bergeron et al. Probing Hyperbolic and Surface Phonon-Polaritons in 2D Materials Using Raman Spectroscopy, *Nat. Commun.* 14, 4098 (2023).

Due to its relevance, both in terms of the Reststrahlen position, MIR modes, and performance metrics, I suggest adding this reference.

Reviewer #2 (Remarks to the Author):

The authors addressed my questions and concerns successfully. I would therefore recommend publication in *Nature*.

Reviewer #3 (Remarks to the Author):

I thank the authors for their detailed attention and positive responses to the reviewer comments. From my perspective I believe that the authors have adequately addressed the concerns raised and I have no further comments. The changes have improved the manuscript and I believe it is suitable for acceptance.

Response to the reviewers' comments

Reviewer #1 (Remarks to the Author):

Highly confined epsilon-near-zero- and surface-phonon polaritons in SrTiO₃ membranes
Ruijuan Xu et al.

The authors have done a good job in clarifying or correcting a number of issues. I have one final comment. The authors wrote in their reply,

“To the best of our knowledge, the TMD materials did not show so far convincing mid- and far-infrared phonon-polaritonic performances (even though they demonstrate nice exciton polaritonic behavior in the near-infrared and the visible range, which is not the topic of this paper).”

I would like to bring to the attention of the authors this following work:

A. Bergeron et al. Probing Hyperbolic and Surface Phonon-Polaritons in 2D Materials Using Raman Spectroscopy, Nat. Commun. 14, 4098 (2023).

Due to its relevance, both in terms of the Reststrahlen position, MIR modes, and performance metrics, I suggest adding this reference.

We thank the Reviewer for the kind comment. We added this reference.

Reviewer #2 (Remarks to the Author):

The authors addressed my questions and concerns successfully. I would therefore recommend publication in Nature.

We thank the Reviewer for the kind comment.

Reviewer #3 (Remarks to the Author):

I thank the authors for their detailed attention and positive responses to the reviewer comments. From my perspective I believe that the authors have adequately addressed the concerns raised and I have no further comments. The changes have improved the manuscript and I believe it is suitable for acceptance.

We highly appreciate the comment from the Reviewer.